# A systems biology approach uncovers cell-specific gene regulatory effects of genetic associations in multiple sclerosis

International Multiple Sclerosis Genetics Consortium

Genome-wide association studies (GWAS) have identified more than 50,000 unique associations with common human traits. While this represents a substantial step forward, establishing the biology underlying these associations has proven extremely difficult. Even determining which cell types and which particular gene(s) are relevant continues to be a challenge. Here, we conduct a cell-specific pathway analysis of the latest GWAS in multiple sclerosis (MS), which had analyzed a total of 47,351 cases and 68,284 healthy controls and found more than 200 non-MHC genome-wide associations. Our analysis identifies pan immune cell as well as cell-specific susceptibility genes in T cells, B cells and monocytes. Finally, genotype-level data from 2,370 patients and 412 controls is used to compute intra-individual and cell-specific susceptibility pathways that offer a biological interpretation of the individual genetic risk to MS. This approach could be adopted in any other complex trait for which genome-wide data is available.

A full list of authors and their affiliations appears at the end of the paper. Correspondence and requests for materials should be addressed to S.E.B. (email: Sergio.Baranzini@ucsf.edu)

Translating Genome-wide association studies (GWAS) discoveries into functionally relevant biology has proven to be highly challenging. The extensive linkage disequilibrium (LD) which typically flanks common variants means that most GWAS identified SNPs are likely to be tags for functionally relevant variation rather than exerting any meaningful effects themselves. Furthermore, since the vast majority of associations identified by GWAS map to non-coding regulatory regions it is likely that the underlying functionally relevant variants only exert pertinent effects on gene expression in particular tissues[1–4]. Fortunately, better powered studies, have increased the number of associations identified enabling biological meaning to be investigated in aggregate (i.e. pathway analysis). In its simplest form, genes lying closest to the most strongly associated (lead) SNP identified for each association can be grouped into pathways or specific functional memberships via the use of pre-assembled controlled vocabularies (Gene ontology, KEGG, etc)[5–8]. This approach can be enhanced by using protein interaction networks to more rigorously assess which of the candidate genes encode proteins that physically interact in any particular pathway[9]. Using this refined approach, we and others have been able to show that MS-associated genes are indeed more likely to interact in protein space[10–12]. Furthermore, this analysis can be extended to include association signals below the genome-wide threshold of significance and thereby nominate new additional potentially meaningful associations[11,13,14]. However, the networks of genes/proteins identified by these approaches are not cell- or tissue-specific, thus limiting the usefulness and interpretability of this information.

With the completion of efforts like the Encyclopedia of DNA elements (ENCODE) and the Roadmap Epigenomics Project (REP) a wealth of information on regulatory elements is now available from hundreds of cell types and dozens of different tissues[15,16], raising the possibility of applying network-based approaches in a cell-specific manner. We reasoned this approach would likely be highly informative in diseases like multiple sclerosis (MS) where substantial numbers of associated variants have been identified. MS is an autoimmune disease of the central nervous system (CNS) and leads to a neurodegenerative process. Our recent GWAS meta-analysis and follow-up study have revealed a total of 233 genome-wide significant associations and a further 416 variants potentially associated with MS[17].

Here we develop a framework to interpret such associations in the context of cell-specific protein networks to identify the most likely process(es) affected by the non-MHC associations as a whole. This approach involves (1) selecting independently associated signals in extended haplotypic blocks; (2) identifying the genomic regulatory processes likely to be altered by the polymorphisms in these blocks in a cell-specific manner; (3) computing a cell-specific gene score for genes in each associated locus; (4) building cell-specific gene/protein networks; (5) Interpreting the biological processes most likely affected for each of the cell types studied. We demonstrate this approach in the latest GWAS meta-analysis in MS involving a total of 47,351 cases and 68,284 controls. Furthermore, we use genotype-level data from a subset of 2370 cases and 412 controls to identify cell-specific intra-individual risk pathways. These individualized scores can be used as a global risk measure in subsequent associations with more detailed phenotypes.

## Results

### Predicted regulatory effects (PRE) of MS-associated variants.
We integrated genetic association signals from the latest genetic analysis of MS[17] with cell specific information on regulatory elements available from the ENCODE and Epigenomics Roadmap projects (REP) to identify cell-specific networks likely affected by the susceptibility variants (Fig. 1a, Supplementary Fig. 1). We included all genome-wide (GW) significant single nucleotide polymorphisms (SNPs) together with their proxies selected at differing LD thresholds ($r^2 > 0.5$ was used for the main analysis).

All regulatory information retrieved for each MS-associated region was compiled for each cell type in a single master table (Supplementary Data 1). This catalogue contains all of the available regulatory features that potentially modulate the expression of each of the genes mapping to each associated region in a cell-specific manner. We next used this information to build a cell-specific, genetic regulatory network that constituted the basis of a gene prioritization scheme within each associated region (Fig. 1b). Specifically, each gene within a given locus received a score (PRE) that does not depend solely on the closest SNP, but that is equal to the weighted sum of all regulatory features potentially affected by variation at nearby associated SNPs (See Methods). Figure 1c shows a heat map representation of the PREs for all genes ($n = 2,444$) implicated by the GW significant MS associations for each of the main cell/tissue types (Individual PREs for each cell/tissue and GWAS statistical confidence are listed in Supplementary Data 2-28). Two GW significant regions (10 and 21) are shown in larger detail as representative examples (Fig. 1c). Because it integrates actual regulatory information for each associated SNP and those in LD, this approach can prioritize the most likely genes affected by the same association signals in each cell type analyzed (Supplementary Data 29). In the example, the PREs at the top of Fig. 1c highlight the lead variant defining Region 10 (rs6670198, Chr 1) and those in LD which are likely to affect the expression of FAM213B and TNFRSF14 in all immune cells (B, T and monocytes -M-). These SNPs would have almost no effect at all in the CNS and lungs (L), as MS-associated variants are unlikely to alter their expression in those tissues. For comparison, a simple proximity approach would have just implicated the FAM213B gene, which, while being of biological interest, may describe an incomplete scenario. Interestingly, TNFRSF14 (a member of the TNF receptor superfamily) encodes a protein involved in signal transduction pathways that activate both inflammatory and inhibitory T-cell immune responses due to its particular ability to interact with multiple ligands in distinct configurations[18].

Another example is provided by region 21, defined by the lead SNP rs6032662 mapping to chromosome 20. rs6032662 maps midway between NCOA5, a tumor suppressor gene, and CD40, which encodes a well-known member of the TNF-receptor superfamily. This receptor is essential in mediating a broad variety of immune and inflammatory responses including T cell activation, T cell-dependent immunoglobulin class switching, memory B cell development, and germinal center formation[19]. While its expression is highest among antigen presenting cells (including those derived from dendritic cells and monocyte/macrophages) its PRE score is low in the M set (Fig. 1c), indicating that, of the cells studied, MS-associated variants only regulate its expression in B cells. Altogether, these results show that this approach is a useful strategy to prioritize genes within association regions.

### Protein connectivity among products of MS-associated loci.
Previous studies have shown that the proteins encoded by genes affected by genetic association signals are more likely to interact, in part because they often participate in the same biological pathways[6,20–22]. Thus, we evaluated (for each cell/tissue) how many of the gene products in MS-risk loci predicted to be positively regulated (i.e. PRE > 25th percentile or PRE-25) also

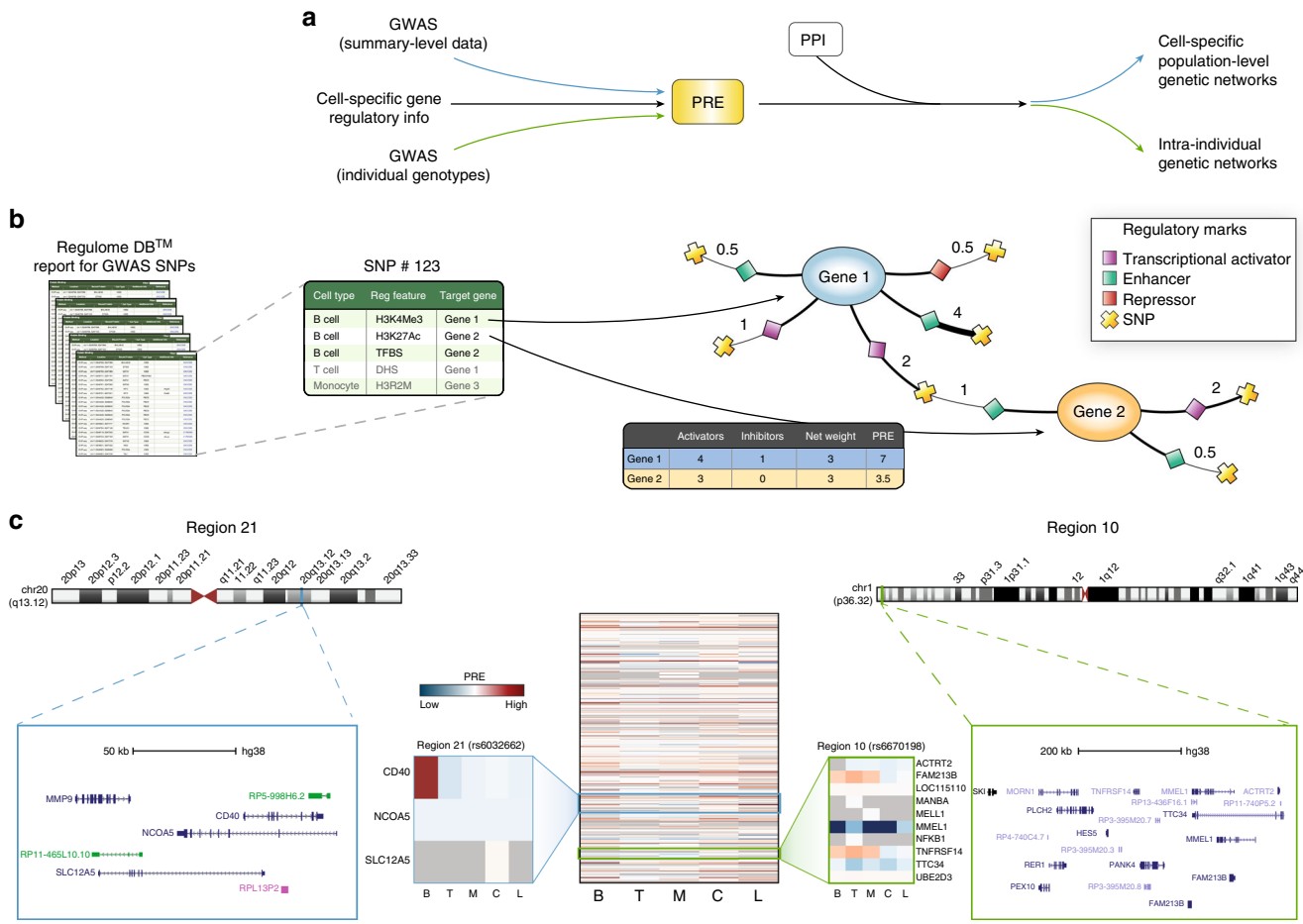

**Fig. 1** Overall strategy and computation of the predicted regulatory effect (PRE) in MS-associated loci. **a** GWAS signals were integrated with cell-specific regulatory information to compute PRE at both population and individual level. In a second stage, genes with high PRE at each of the cell types analyzed were identified in a human protein interactome (PPI) and sub-networks of enriched genes (proteins) were extracted. **b** Each MS-associated SNP and those in LD were used as query in RegulomeDB. For each SNP, the all regulatory features were annotated and classified according to type and cell of origin. A graph connecting every queried SNP (crosses), the regulatory feature (diamonds), and the target gene (circles) was created and the number of experiments supporting a particular regulatory feature was used as weight (numbers next to SNP). Finally, a PRE score was computed for each gene by summing up weights from all incoming regulatory signals for each of the cell types analyzed. **c** Heatmap represents the PRE of all genes under GW MS-associated loci for cells of interest. Rows represent genes, and columns denote cell types. Colors indicate positive (red), neutral (white) and negative (blue) PRE values. Two representative regions are highlighted. Region 10 (associated SNP: rs6670198, green box) highlights immune-specific (B, T, and M) regulation of *FAM213B* and *TNFRSF14*. In contrast, region 21 (associated SNP rs6032662, blue box), shows high PRE only for *CD40* in B cells. C: CNS; L: lung; T: T cells; M: monocytes; B: B cells. This analysis represents all SNPs with an $r^2 > 0.5$ of the main GW effect

interacted in a human protein network containing 15,783 proteins and 455,321 interactions (See methods) (Fig. 2 shows a schematic of this approach). In addition to the total number of interactions we also computed other relevant network metrics such as the size of the largest connected component –LCC– and the number of connections –edges- among nodes within the LCC. These metrics, if statistically significant, are often good indicators of true biological networks. We found that, for all immune cell types analyzed, the above network metrics among GW associated gene products always exceeded those from 10,000 randomly generated, size-matched networks. Specifically, network metrics among the gene products of GW associated loci were statistically significant for T cells, B cells and monocytes (Fig. 3). The number of interactions among genes related to the CNS were not significantly different than expected. One factor likely affecting this result is that in contrast to immune cells (for which PREs were computed on a cell-specific manner) computations for CNS are the result of 25 different cell types/anatomical regions. This could potentially smooth the overall estimate of PREs as the various cell types within the CNS could be under different regulatory control.

The three significant networks (Fig. 3a–c) shared several of their genes, thus constituting a core module (Fig. 3d). The molecular functions of most of these genes belonged to the *binding* (36%) and *catalytic activity* (33%) categories. Other functions were *receptor (13%)*, *signal transduction (6%)*, and *structural molecule (10%)*. A PANTHER analysis revealed these genes belong to JAK/STAT, IFNgamma, interleukin, and integrin signaling pathways, among others. Altogether, our analyses suggest that susceptibility to MS stems from a core of processes that can be active in any of several immune-related cell types. These findings are in agreement with the "omnigenic" model of inheritance of complex diseases, posing that gene regulatory networks are sufficiently interconnected such that all genes expressed in disease-relevant cells are liable to affect the functions of core disease-related genes[23].

As predicted by the omnigenic model, we noted that several genes were only present in some cell types but not others. For example, *CD28* was only present in the T cell network, *ELMO1* in B cells and *MERTK* in the monocyte/macrophage lineage. *CD28* is located on the surface of T cells and provides a required

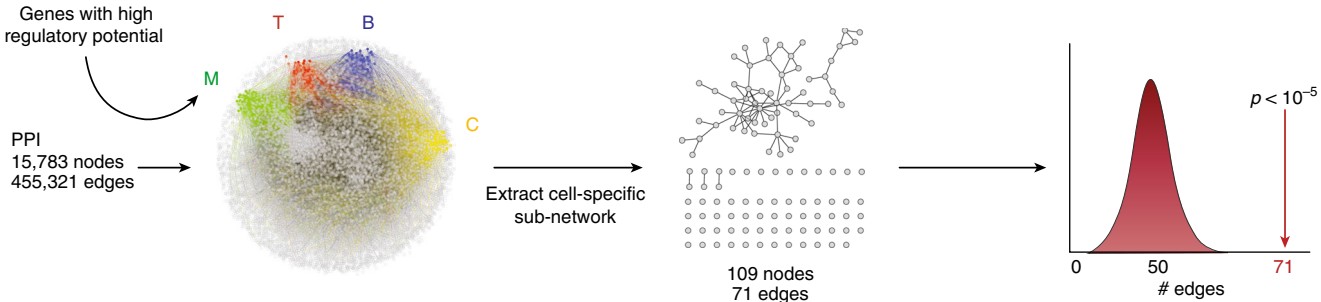

**Fig. 2** Network connectivity analysis. The PRE of genes were loaded as attributes in a protein interactome. In the central panel, genes with a PRE above the 95th percentile of their respective cell-specific distributions are visualized (M: monocyte, green; T: T cells, red; B: B cells, blue, C: CNS, yellow). For each cell type, the number of edges in the sub-network composed of interacting proteins with PRE above the threshold was analyzed. In this example, the CNS sub-network is composed of 109 nodes and 71 edges. Ten thousand random networks with the same number of nodes (i.e. 109) were generated and the distribution of edges was plotted along with the number of edges of the relevant sub-network (i.e. 71). A *p*-value was computed to evaluate the probability that this number of edges was seen by chance

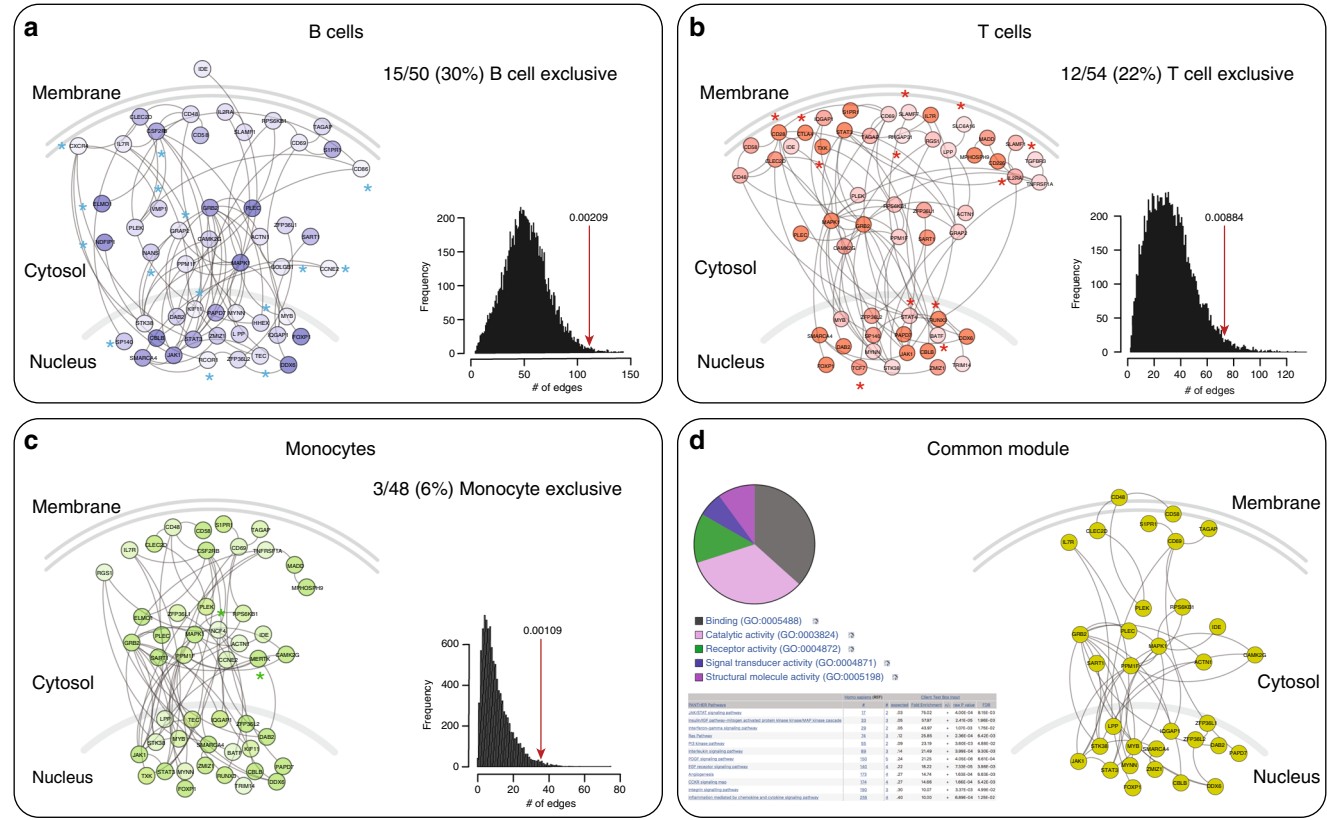

**Fig. 3** Cell-specific gene sub-networks of GW associated regions ($r^2 > 0.5$). Graphs correspond to the largest connected component in each cell/tissue bucket. Nodes represent proteins and edges represent interactions. For each cell type the PRE is proportional to the color intensity (dark: high; light: low). Genes/proteins are organized according to their cellular distribution. The histogram next to each sub-network shows the distribution of the number of edges of 10,000 randomly generated networks. The red arrows denote the number of edges observed in the corresponding sub-network and the *p*-value, the probability of observing a more extreme number of edges in a randomly generated network. **a** B cells; **b** T cells; **c** monocytes. An asterisk is placed next to genes/proteins exclusively observed in that cell type. **d** shows an aggregate (common) module present in all three cell types. A pie chart describes the GO: molecular functions assigned to these genes and a table describes the nine PANTHER pathways that were significantly enriched

co-stimulatory signal to trigger their activation after engagement of the MHC-antigen-T cell receptor trimolecular complex[24]. *ELMO1* encodes a cytoplasmic adaptor protein that interacts with DOCK family guanine nucleotide exchange factors to promote activation of the small GTPase RAC, thus enabling lymphocyte migration[25]. The *MERTK* gene encodes a receptor tyrosine kinase

that transduces signals from the extracellular matrix into the cytoplasm regulating many physiological processes including cell survival, migration, differentiation, and phagocytosis of apoptotic cells (efferocytosis). Specifically, *MERTK* plays several important roles in normal macrophage physiology, including regulation of cytokine secretion and clearance of apoptotic cells[26]. These

observations suggest that, in addition to the core susceptibility module, at least part of the risk is cell type-specific.

We next performed a sensitivity analysis by testing separately GW, statistically replicated (SR), and non-replicated (NR) effects (see Methods) SNP sets at three different LD cut-offs ($r^2 > 0.1$, $r^2 > 0.5$, and $r^2 > 0.8$) and three PRE thresholds (PRE-10, PRE-25 and PRE-50) (Supplementary Data 30). As anticipated, although the most significant network metrics were seen in the analysis based on GW signals (Supplementary Fig. 2, panel A), some significant network metrics were also seen in the analysis based on the SR SNPs (Supplementary Fig. 2, panel B). This confirms that in well powered studies variants with evidence for association just short of GW significance may still represent real effects[14]. In contrast, the connectivity of networks obtained with SNPs from the NR set was usually not significantly higher than that of random networks (Supplementary Fig. 2, panel C). As expected, including the wider range of SNPs implicated by relaxing the LD threshold down to $r^2 > 0.1$ resulted in less significant network metrics, suggesting that including less robust proxies introduced more noise than signal.

**Individualized PRE correlate with gene expression**. The detailed mapping of regulatory information for each SNP suggests that if PRE are computed for a given cell type in a single individual based on the carriage of relevant risk alleles, these values should capture a non-negligible proportion of the variance in gene expression in that cell type. To test this hypothesis, we interrogated the expression of the entire transcriptome of FACS-sorted $CD4^+$ T cells, and $CD14^+$ monocytes from 25 MS patients by RNAseq and then assessed the correlation of their genotype dependent PRE and their actual gene expression in each cell type separately. Our results showed that the correlation observed was in all cases significantly higher than what would be expected by chance if these metrics were independent. Furthermore, the computed correlations were always higher for the matching cell type (CD4/CD8 expression with T cells PRE and CD14 expression with monocytes PRE) (Table 1). The average correlation between RNA expression and PRE within the same cell type was 0.331 (CD4 vs. T cells, $p < 10^{-300}$, linear regression), 0.324 (CD8 vs. T cells, $p < 10^{-300}$, linear regression), and 0.246 (CD14 vs. monocytes, $p < 10^{-300}$, linear regression), representing a significantly higher than expected value for each cell type. Correlations between PRE and RNA expression of mismatched cell types were significantly lower. These results suggest that the computation of PRE can be applied to single patients and individual scores can be generated for each of them.

We then used the genotype-level data from one of the GWAS datasets (UCSF, Supplementary Data 31), composed of 2370 patients and 412 controls, to compute cell-specific risk scores for each individual using the same pipeline used for the population-level data. Rather than considering all 200 associations, this personalized approach takes into account the specific risk alleles present in each individual and thus enables exploration of subject heterogeneity in a biological context and in a cell specific manner. Hierarchical clustering of subjects and heatmap visualizations of the PRE of genes under regions 9, 49, and 53 ($r^2 > 0.5$) for all cell types in this subset of cases and controls are shown as an example (Fig. 4). The heterogeneity across individuals in the PRE of a given gene can be readily seen for all cell types. As expected for common variants, cases and controls (denoted by red and green horizontal bars in the leftmost column) do not cluster separately within any single association region. This analysis provides a visual representation of which genes are most likely affected by common variants associated with the disease in those individuals, in a cell-specific manner. The PRE for most genes in each heatmap show ample variability across individuals, highlighting genetic differences in their susceptibility to MS at this locus. These maps also reveal which genes are most likely to be affected in each cell type by these common alleles. For example, while associated variants near the gene *EOMES* (region 9) potentially modulate its expression in T cells, this gene is less regulated in B cells and monocytes and strongly silenced in CNS (Fig. 4a). This is consistent with its function as a critical transcription factor in T cell differentiation[27,28]. Interestingly, higher PRE of these variants are observed for Th2 cells than for any of the other subsets analyzed. Previous reports revealed that *EOMES* expression limits *FOXP3* induction, thus effectively reducing Treg populations[29]. Genomic variation at this locus resulting in dysregulated *EOMES* expression in T cells and NK cells might be a critical mediator of the risk to MS[30].

Another interesting example is the high PRE observed for the gene *CD40* (region 21) preferentially in B cells for most subjects (Fig. 4b, pink boxes). This finding is consistent with the critical role of *CD40* in B cell development and maturation, and indicates that MS risk affecting B cell biology is higher in some subjects than others. Furthermore, individuals carrying the risk variants within *CD40* (rs4810485*T) have been reported to express lower levels of *CD40* in the surface of their B cells and lower IL-10 levels[31]. This could carry therapeutic implications considering the prominent role of B cell depletion therapy in this disease[32,33]. Another signal revealing B cell involvement in MS risk is the high PRE of all members of the Fc receptor-like (*FCRL*) gene family in region 53 (Fig. 4c, yellow boxes) preferentially in B cells for most subjects, consistent with the function of these gene family as regulators of proliferation in B cells and phagocytosis[34].

**Intra-individual risk networks are more connected in MS**. Finally, we integrated individual risk regulatory scores with a global protein interactome to compute intra individual, cell specific risk networks. We hypothesize that these networks would provide a step forward in the description of aggregate personalized risk scores by representing risk in a biologically relevant manner[35,36]. Furthermore, building risk profiles with pathway and cell specific information describes more accurately the biology potentially affected by risk variants inherited by a particular individual. We also hypothesized that similarly to what we observed for cases and controls at the population level, more interactions among proteins encoded by risk loci would be observed for cases than for controls. This was indeed the case, as we observed statistically significant differences between cases and controls for the three main cell types tested (the CNS was not significant as shown in Fig. 5a and Supplementary Fig. 3) (Supplementary Data 32). The largest number of intra-individual interactions among gene products with high PRE was observed in monocytes, followed by T cells, B cells and the CNS (Fig. 5a

---

**Table 1 Correlation between regulatory potential and cell specific global gene expression**

| PRE | CD4 $r$ ($p$-value) | CD8 $r$ ($p$-value) | CD14 $r$ ($p$-value) |
|---|---|---|---|
| Monocyte | 0.225 ($p < 10^{-300}$) | 0.218 ($p < 10^{-314}$) | **0.246** ($p < 10^{-300}$) |
| T-cell | **0.331** ($p < 10^{-300}$) | **0.324** ($p < 10^{-300}$) | 0.253 ($p < 10^{-300}$) |
| B-cell | 0.204 ($p < 10^{-248}$) | 0.204 ($p < 10^{-247}$) | 0.219 ($p < 10^{-287}$) |
| CNS | 0.108 ($p < 10^{-75}$) | 0.112 ($p < 10^{-82}$) | 0.120 ($p < 10^{-93}$) |
| Lung | 0.117 ($p < 10^{-86}$) | 0.122 ($p < 10^{-94}$) | 0.133 ($p < 10^{-112}$) |

Bold values indicate the corresponding cell type for the RNAseq results (e.g. CD14 RNAseq data corresponds to monocytes, and CD4 and CD8 corresponds to T cells)

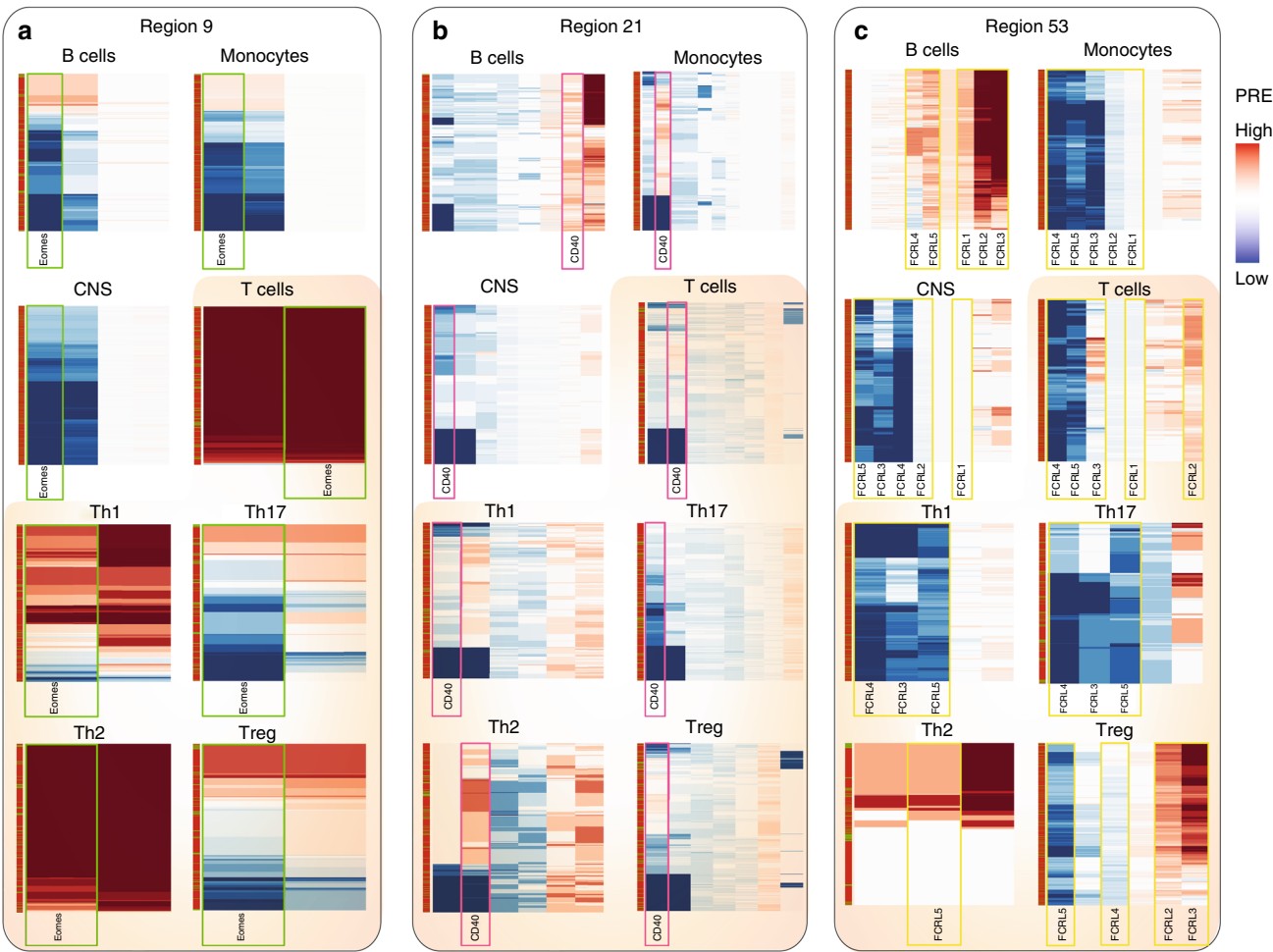

**Fig. 4** Individualized PRE computations for three representative associated regions. Each row represents an individual (out of 2370 cases and 412 controls), and each column represents a gene within the associated region. Region 9 (**a**) contains the gene *EOMES* (green boxes), region 21 includes *CD40* (pink boxes) (**b**) and region 53 (**c**) the FC receptor-like cluster (yellow boxes). The leftmost column denotes subject status (red: cases; green: controls)

shows results for PRE-25). This is consistent with the larger significance of these risk networks observed at the population level (Fig. 3). Interestingly, PRE values correlate with the global polygenic risk (Supplementary Fig. 4) but it uniquely enables identification of high-risk and low-risk individuals in a cell-specific manner. Figure 5b highlights four case:control pairs with different risk profiles in each of the four main cell types/tissues analyzed. For example, subject 201100870 (case) is at the 99th percentile of the distribution of network edges for monocytes (120 nodes and 271 edges). In contrast, subject 20020214 (control) is at the 1st percentile (61 nodes and 98 edges).

Another interesting observation emerging from this analysis is that subjects at the extremes of the distribution of intra-individual interactions (a proxy for their overall risk) can be identified for each cell type. For example, subject 201327986 has 110 nodes and 190 edges in this subject's B cell risk network (blue box), corresponding to the 99th percentile of all cases (Fig. 6). In contrast, the corresponding percentile of the number of edges in his T, M and C (red/green/yellow) networks is substantially lower (47th/49th/66th). In line with results observed at the population level, individual CNS risk networks are consistently smaller and less connected than those from B cells, T cells and monocytes. Although CNS is still the least connected network in subject 201101897, with 118 edges in its CNS risk network (yellow box), it ranks in the 99th percentile of all cases. In contrast the

percentile connectivity of T cell (51st), B cell (29th) and M (75th) risk networks for this subject rank noticeably lower.

While the number of interactions (edges) among proteins encoded by genes in associated loci was variable across cell types, on average, more interactions were observed for cases than for controls in all cell types studied. Supplementary Fig. 3 shows the significance of testing different network parameters between cases and controls across a wide range of conditions. Similar to what we observed at the population level, significant effects were also seen for loci with less than genome-wide evidence for association, suggesting that some of those variants considered less strongly significant, also confer risk.

In summary, this analysis underscores the notion that total MS risk is not only carried by accumulation of risk alleles, but also by how the genes and proteins affected by those polymorphisms interact within each cell type. We anticipate this model could apply to other common diseases.

## Discussion

In this work, we provide evidence that integration of associated variants from GWAS with regulatory information and protein interactions, provide plausible models of disease pathogenesis. This approach not only offers a data-driven solution to prioritize which genes within a locus are most likely affected by the risk

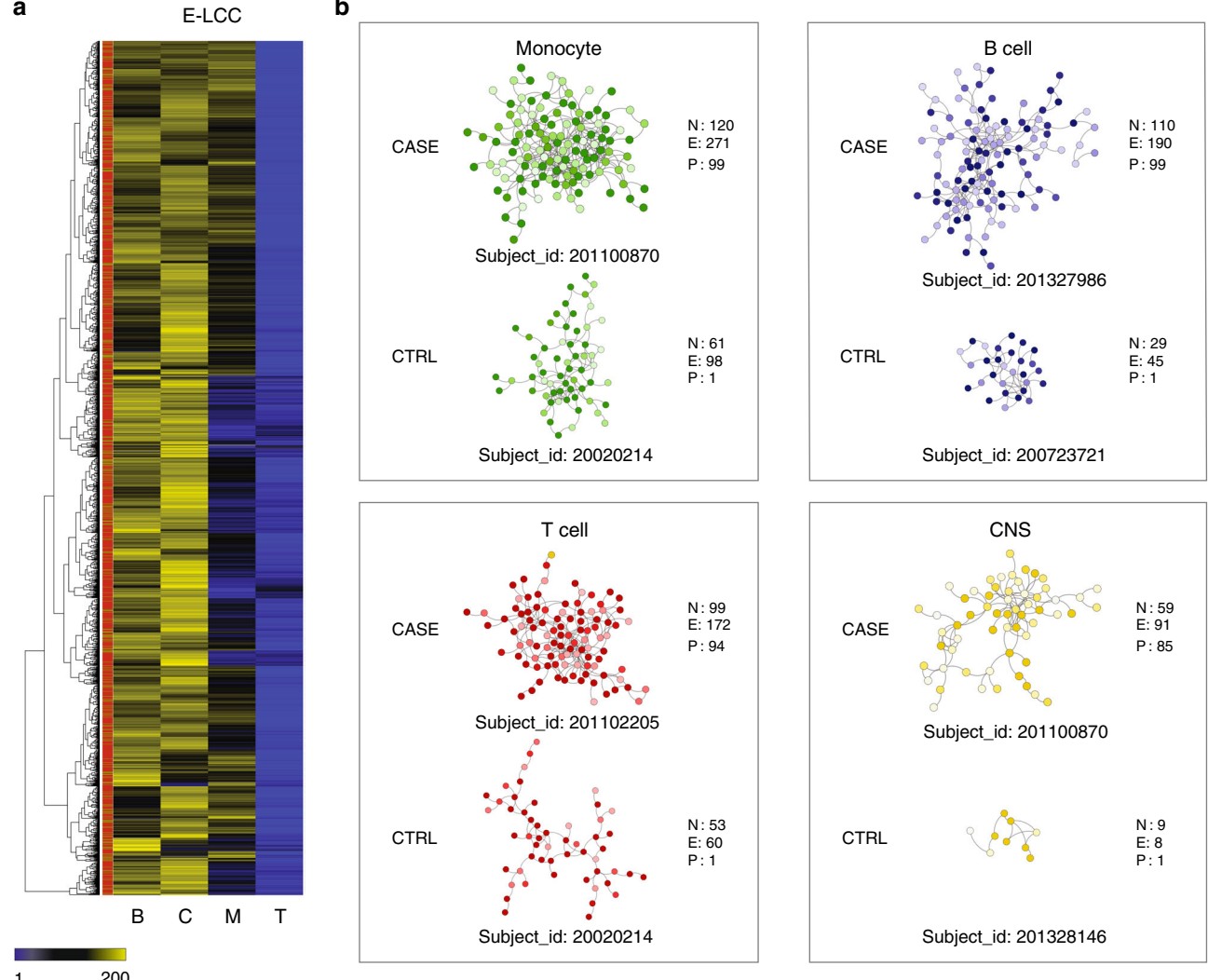

**Fig. 5** Select case-control intra-individual MS-risk networks. **a** Number of edges in the largest connected component (LCC) of the network generated among proteins (genes) with high PRE (>25th percentile) in 2370 patients and 412 healthy controls (GW_$r^2$ > 0.5). Each row represents a subject, each column represents a cell type (B: B cell; T: T cell; M: monocyte; c: CNS). The leftmost column indicates subject status (red: cases; green: controls). **b** Representative sub-networks from subjects at the extremes of the distribution for E-LCC for each cell type. For each network, the number of nodes (N), edges (E), and percentile relative to all subjects (P) is indicated. The intensity of node color is proportional to the PRE of each gene in the corresponding cell type

variant(s) but also provides an interpretable model of risk in a cell/tissue specific manner. While the PRE scores computed here are significantly correlated with actual gene expression from the corresponding cells, the correlation is partial, thus underscoring a potential limitation in our approach, and of available data. However, the statistically significant results obtained for genes and pathways known to be involved in MS further validates this approach. Indeed, the models presented here are consistent with MS genetic risk being driven by the long-term alteration of cellular pathways primarily in monocytes, but also in the B and T cell arms of the human adaptive immune response. The smaller but not negligible contribution of CNS pathways to MS risk is in agreement with our previous analysis[17], which identified the monocyte/macrophage/microglia axis as a key player in directing the autoimmune process to the CNS. However, an important caveat, particularly affecting results for this compartment, must be taken into account: for this analysis, the CNS group was composed of a heterogeneous ensemble of purified primary cells, established cell lines and dissected specimens from specific

anatomical regions as available in the ENCODE and REP datasets. Although all derived from CNS tissue, it is highly likely that different regulatory mechanisms are at play in each cell type (in some cases resulting in drastically different expression patterns), thus somehow confounding the overall CNS signature computed here. Thus, the detected effect of MS-associations which map to the CNS could represent the lower boundary of a more widespread phenomenon. A more detailed CNS-specific data set containing genome-wide regulatory element information might be needed to address this question in larger detail.

In the last few years, several post-GWAS pathway approaches such as DEPICT[37], FUMA[38] and PASCAL[39] have been proposed and utilized to interpret and integrate summary statistics into a biologically meaningful model. While sharing some of the basic characteristics of previous approaches, our method features a set of unique properties, most notably the introduction of data-driven regulatory effects of associated variants (and those in LD), the ability to create cell-specific networks, and the computing of individual disease burden maps (Supplementary Table 1). Given

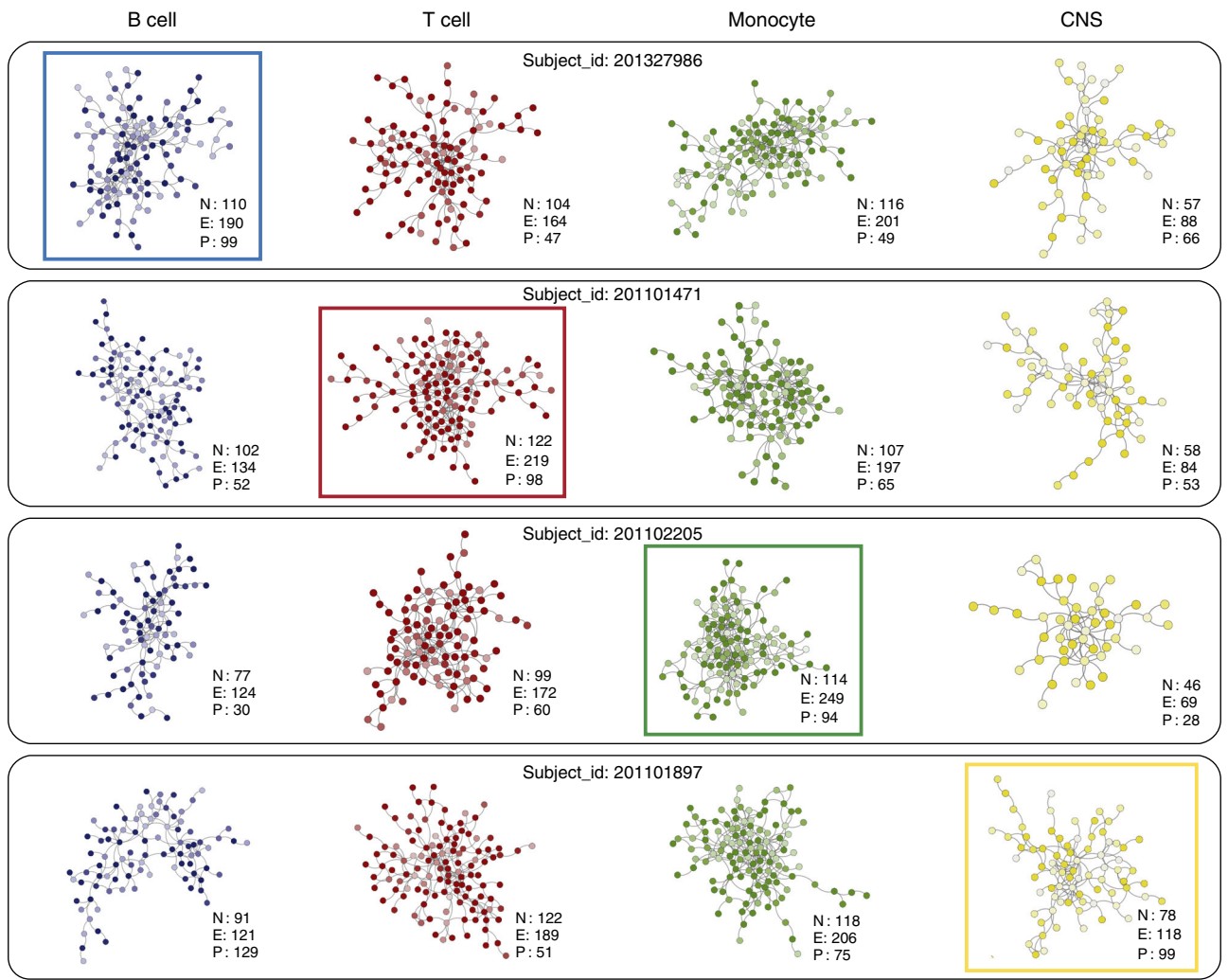

**Fig. 6** Heterogeneity in intraindividual MS-risk networks Intraindividual cell-specific networks of four representative MS subjects showing heterogeneity of risk across all cell types. **a** Cell specific risk networks for subject_id: 201327986. **b** Cell specific risk networks for subject_id: 201101471. **c** Cell specific risk networks for subject_id: 201102205. **d** Cell specific risk networks for subject_id: 201101897. For each subject, the most connected risk network (number of edges in the highest percentile across all subjects) is highlighted within a colored box. For each network, the number of nodes (N), edges (E), and percentile relative to all subjects (P) is indicated. The intensity of node color is proportional to the PRE of each gene in the corresponding cell type. M: monocyte, green; T: T cells, red; B: B cells, blue, C: CNS, yellow

that each method produces a different output, it is not possible to directly compare these approaches. However, the first and arguably most important step in all these tools (including ours) is to compute the SNP to gene values. A basic comparison of these three tools shows that the exact same genes were prioritized for almost half of all non-HLA associated loci (97/200). A closer look revealed that our method was the only one that called at least one gene per associated locus, and produced gene prioritization sets with the least ambiguities (Supplementary Table 2 and Supplementary Fig. 5).

Recent evidence has emerged that polygenic risk scores for schizophrenia associate with therapeutic response to Lithium-based therapies[40]. Similar approaches are being tested for other psychiatric, oncological and cardiovascular diseases[41–45]. The observation that some risk variants only affect expression of a given gene in one cell type but not in others, may at least in part, underlie the observed clinical heterogeneity in the MS population. Thus, when this approach is implemented at the individual level, specific risk profiles can be built for each subject with MS. We

speculate that in the near future this information could also be used as the basis to develop individualized risk scores, or to derive personalized approaches to therapy. For example, a subject with high B cell genetic risk may be a good candidate for B cell depletion therapies, while a subject with a high T cell risk may benefit the most from immunomodulatory drugs that target T cell function or migration into the CNS.

This cell-specific pathway approach can be extended to any set of SNPs of interest in any condition at both population (summary) and individual (genotype) levels.

## Methods

**Predicted regulatory effects (PRE)**. Genome-wide regulatory elements from ENCODE and REP were collected from regulomeDB[46] (which contains more than 400 million genomic regulatory features collected from 400 cell and tissues; Supplementary Table 3) for all non-MHC independent effects (SNPs)[17]. Specifically, single nucleotide polymorphisms (SNP) corresponding to all non-MHC GW ($n = 200$; Supplementary Data 33), SR ($n = 416$; Supplementary Data 34) and NR ($n = 3695$; Supplementary Data 35) were extracted for analysis GW, SR, and NR as defined previously[17]. The 200 GW effects were distributed in 156 unique regions

(44 regions contained multiple independent effects). Similarly, the 416 SR effects were distributed in 354 unique regions (62 regions contained multiple independent effects) and the 3695 NR effects were distributed among 1,883 unique regions (1812 regions contained multiple independent affects). Three sets of SNPs were created for each region according to their $r^2$ with their corresponding main effect ($r^2 \geq 0.8$, $r^2 \geq 0.5$, and $r^2 \geq 0.1$).

A python tool was written to automatically fetch data from RegulomeDB for these SNPs in all three lists (totaling 538,826 SNPs). Similarly, data for SNPs in different levels of LD ($r^2 > 0.8$, $r^2 > 0.5$, and $r^2 > 0.1$) with each primary effect were also retrieved using chromosomal positions. The main analysis was performed using $r^2 \geq 0.5$ whereas the other sets were only used for the sensitivity analysis. In total, 538,826 SNPs were included in the analysis.

To investigate the effect of SNPs across different cell types and to assess which gene has most potential of being regulated across various cell types, we grouped the cell types present in ENCODE (Supplementary Table 4) and REP (Supplementary Table 5) into four major cell types (buckets). Specifically, these were B cells, T cells, CNS (central nervous system), and M (monocytes). T cell subsets (Th1, Th2, Th17, and Treg) were also analyzed as a separate group. We also built a dataset from lung (L, a cell type/tissue not considered to play a major role in MS susceptibility) as a control. Cancer cell lines were excluded for this analysis.

Regulatory elements were grouped into two major classes: PEX (promoter/enhancer/activator) and R (repressors/inactivators). Cell or tissue of origin was recorded for each regulatory feature, and cell-specific information was grouped into three main cell types (B cells, T cells, monocytes) and one tissue (CNS) that are of interest in MS. In total, 25 brain regions were considered (15 from ENCODE and 10 from REP). In addition, T cell subsets deemed relevant in the pathogenesis of MS (Th1, Th2, Th17, and Treg) were also analyzed separately. Primary cells and cell lines from a tissue not known to be involved in MS (lung, L) were included as control. In addition, eQTL data for T cells and monocytes from the IMMVAR project[47] were integrated into the PRE computations. The HTML data from the scrapped output was parsed to populate data present in regulomeDB tables. A master table was then compiled with each field of regulomeDB data for all 538,826 SNPs.

Multiple regulatory features were considered including protein binding, transcription factor binding sites (TFBS), promoters, enhancers, insulators, histone modifications, and DNAse hypersensitive regions (DHS). We classified these into three broad groups representing promoter/enhancer/transcription (PEX), inert/quiescent (ZQI), and repressor (R, Supplementary Table 6). We next computed weighted SNP-based scores based on the genotype and number of risk alleles to quantitate the regulatory influence of variation at each SNP. The weights were counted as positive if there was evidence that the region promotes transcription and as negative if there was evidence of repression. The weights were then normalized by the total number of experiments conducted for each respective cell type to remove bias against well-studied cell types. These weighted weights (WW) were summed up across SNPs resulting in a sum of weighted weights (SWW, or predicted regulatory effect -PRE-) per gene per region in each cell type. The WW concept derives from the fact that the sum of the effects of neighboring SNP to a given gene is weighted twice. The first time we weight the number of experiments reported in ENCODE or REP for a given SNP-gene pair (e.g. we assign more value to a relationship that has been reported in 10 independent experiments, to another that has been reported just once). The second time, we weight the evidence stemming from all SNPs nearby a gene (depending on the LD structure there could be ~100 SNP near a given gene). A gene with a positive score indicates there is evidence that the region containing the MS-associated SNP(s) is actively influencing its transcription in that particular cell type and vice-versa.

All computations were performed in parallel using the 7400-core QB3 computer cluster at UCSF.

### Protein interaction network-based pathway analysis (PINBPA).
An experimentally determined human protein interactome consisting of 15,783 nodes and 455,321 edges was used for this part of the analysis[21]. We loaded the network into Cytoscape[48] and created cell-specific sub-networks using gene expression values from elsewhere. Specifically, we filtered interactions realized only by gene products expressed in a given cell type, by using RNAseq expression profiles from Kitsak et al.[49]. Thus, for the T cell interactome, we only retrieved interactions between proteins known to be expressed by any T cell subset present in Kitsak et al. In the case of CNS, while gene expression data is sufficiently granular (profiles for different brain cell types and regions exist), epigenomic data for CNS cells/tissues in ENCODE or REP is very sparse, thus we decided to merge all data into a single CNS category.

Next, we loaded the gene-level PRE for each cell type as node attributes and conducted a topological analysis by selecting the subnetwork corresponding to the largest connected component of nodes with positive PRE (those with negative scores are assumed not to be expressed, and thus not to be active players of the interactome). To eliminate noise from very small or loosely unconnected networks, only those with more than 15 nodes were considered. Sensitivity analysis was performed by defining different thresholds on PRE values (10th, 25th, 50th percentiles) and building networks with only proteins exceeding these thresholds (Supplementary Fig S1). Individual network analysis was performed considering differing sets of the potentially associated SNPs identified in our recently completed meta-analysis; those SNP that showed statistically significant evidence of

replication and reached genome-wide significant in the final combined analysis (GW), those that showed statistically significant evidence of replication but did not reach genomewide significance in the final combined analysis (SR) and those failing to show statistically significant evidence of replication (NR). For each cell type, the number of nodes and edges of each subnetwork and that of its largest connected component were computed. The statistical significances were computed by comparison against a background distribution of 10,000 networks of equal size sampled randomly from the same PPI.

### Cell-specific transcriptomes.
This work was approved by the Institutional Review Board at the University of California San Francisco (IRB# 10-00104). PBMCs were obtained from 25 individuals by Ficoll method using Vacutainer CPT tube (BD Biosciences). Subjects were consented according to institutional (UCSF) review board (IRB) guidelines. Three different cell subsets (CD4$^+$ and CD8$^+$ T cells, CD14$^+$ monocyte) were sorted into RLT buffer using a MoFlo Astrios cell sorter (Beckman Coulter). Helper T cells were defined as CD3$^+$CD19$^-$CD4$^+$, cytotoxic T-cell were CD3$^+$CD19$^-$CD8$^+$, and monocytes were sorted as CD14+ cells. Total RNA was isolated from sorted cell subsets using RNeasy Mini kit (Qiagen) and assessed RNA quality using Agilent 2100 Bioanalyzer (Agilent Technologies). 3′ mRNA-Seq libraries for all cell subsets were prepared from 100 ng total RNA using QuantSeq kit (Lexogen) according to the manufacturer's instructions and sequenced 50-bp single-end on the HiSeq 4000 (Illumina). Sequence reads were mapped to the human genome reference (GRCh38) with Gencode annotation (r26) using STAR aligner[50]. Reads were normalized by median of ratios using the DEseq2 package[51]. The R function featureCounts was used to obtain gene-level read counts[52].

We selected overlapping genes between RNA-Seq gene counts and PRE scores, and Pearson's correlation test was performed using the cor.test function in R. The significance of the correlation was confirmed by permutation testing ($n = 1000$).

## Data availability
All data generated or analysed during this study are included in this published article (and its supplementary information files). RNA samples used in this work have been utilized in its entirety and thus are not available. Raw RNAseq (fastq) files used in this work have been deposited in the UCSF Data Share Server [https://doi.org/10.7272/Q6HQ3X3M].

## Code availability
Computer code used in this study is available upon request.

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

## Acknowledgements

This investigation was supported in part by the following sources: NIH/NINDS awards R01NS088155 and 1R01NS099240, the Valhalla Charitable Foundation, and the Heidrich Family and Friends Foundation (Sergio E. Baranzini). US National Multiple Sclerosis Society (TA 3056-A-2), the Harvard NeuroDiscovery Center and an Intel Parallel Computing Center award (Nikolaos A. Patsopoulos). Swedish Medical Research Council; Swedish Research Council for Health, Working Life and Welfare, Knut and Alice Wallenberg Foundation, AFA insurance, Swedish Brain Foundation, the Swedish Association for Persons with Neurological Disabilities. Cambridge NIHR Biomedical Research Centre, UK Medical Research Council (G1100125) and the UK MS society (861/07). NIH/NINDS: R01 NS049477, NIH/NIAID: R01 AI059829, NIH/NIEHS: R01 ES0495103. Research Council of Norway grant 196776 and 240102. NINDS/NIH R01NS088155. Oslo MS association. Research Council KU Leuven, Research Foundation Flanders. AFM, AFM-Généthon, CIC, ARSEP, ANR-10-INBS-01 and ANR-10-IAIHU-06. Research Council KU Leuven, Research Foundation Flanders. Inserm ATIP-Avenir Fellowship and Connect-Talents Award. German Ministry for Education and Research, German Competence Network MS (BMBF KKNMS). Oslo MS association, Research Council of Norway grant 196776 and 240102. Dutch MS Research Foundation. TwinsUK is funded by the Wellcome Trust, Medical Research Council, European Union, the National Institute for Health Research (NIHR)-funded BioResource, Clinical Research Facility and Biomedical Research Centre based at Guy's and St Thomas' NHS Foundation Trust in partnership with King's College London. German Ministry for Education and Research, German Competence Network MS (BMBF KKNMS). Italian Foundation of Multiple Sclerosis (FISM). NMSS (RG 4680A1/1). German Ministry for Education and Research, German Competence Network MS (BMBF KKNMS). Lundbeck Foundation and Benzon Foundation.

## Author contribution

L.F.B., L.B., F.M-B., D.B., M.C., G.C., B.A.C.C., S.D., K.D., P.D., D.E., F.E., B.F., A.G., B.D., G.H., C.H., B.H., R.H., D.H., N.I., S.K., J.K., M.K., I.K., R.M., A.O., A.P., M.P-V., J.S., T.O., B.V.T., G.S., H.F.H., A.C., S.L.H., D.A.H., F.Z., P.D., S.S., and J.R.O. directly contributed to sample acquisition. S.B., J.M., K.K., S.J.C., and S.E.B. performed DNA/RNA processing. A.B., J.M., S.J.C., P.D., and S.E.B. performed genotyping and/or RNAseq. L.M., N.A.P., C.C., S.D.B., A.B., J.M., K.K., X.J., T.F.A., T.B., F.M-B., D.B., F.B., E.G.C., B.A.C.C., C.G., A.G., P-A.G., J.H., N.I., C.M.L., M.L., F.L., R.H., J.S., H.F.H., S. L.H., D.A.H., F.Z., P.D., S.S., J.R.O., and S.E.B. performed analysis. L.M., N.A.P., C.C., A.B., J.M., K.K., A.S., S.J.C., J.H., and S.E.B. we involved in data handling (clinical, demographic or genotypic). N.P., C.C., P-A.G., J.H., B.H., I.K., R.H., A.I., B.V.T., G.S., H.F.H., A.C., S.L.H., D.A.H., F.Z., P.D., S.S., J.R.O., and S.E.B. designed the study. L.M., C.C., S.D.B., J.M., K.K., S.L.H., S.S., J.R.O., and S.E.B. wrote the paper.

## Additional information

**Competing interests:** The authors declare no competing interests.

## International Multiple Sclerosis Genetics Consortium

Lohith Madireddy[1], Nikolaos A. Patsopoulos[2,3], Chris Cotsapas[4,5], Steffan D. Bos[6,7], Ashley Beecham[8], Jacob McCauley[8,9], Kicheol Kim[1], Xiaoming Jia[1], Adam Santaniello[1], Stacy J. Caillier[1], Till F.M. Andlauer[10,11], Lisa F. Barcellos[12], Tone Berge[7,13], Luisa Bernardinelli[14], Filippo Martinelli-Boneschi[15,16], David R. Booth[17], Farren Briggs[18], Elisabeth G. Celius[7,19], Manuel Comabella[20], Giancarlo Comi[21], Bruce A.C. Cree[1], Sandra D'Alfonso[22], Katrina Dedham[23], Pierre Duquette[24], Efthimios Dardiotis[25], Federica Esposito[21], Bertrand Fontaine[26,27], Christiane Gasperi[10], An Goris[28], Bénédicte Dubois[28], Pierre-Antoine Gourraud[29,30], Georgios Hadjigeorgiou[31], Jonathan Haines[18], Clive Hawkins[32,37], Bernhard Hemmer[10], Rogier Hintzen[33,34], Dana Horakova[35], Noriko Isobe[36], Seema Kalra[32,37], Jun-ichi Kira[36], Michael Khalil[38], Ingrid Kockum[39], Christina M. Lill[40,41], Matthew R. Lincoln[4], Felix Luessi[40], Roland Martin[42], Annette Oturai[43], Aarno Palotie[44,45], Margaret A. Pericak-Vance[8,9], Roland Henry[1], Janna Saarela[44], Adrian Ivinson[46], Tomas Olsson[39], Bruce V. Taylor[47], Graeme J. Stewart[17], Hanne F. Harbo[6,7], Alastair Compston[48], Stephen L. Hauser[1], David A. Hafler[4], Frauke Zipp[40], Philip De Jager[3,49], Stephen Sawcer[48], Jorge R. Oksenberg[1] & Sergio E. Baranzini [1,50]

[1]Weill Institute for Neurosciences, Department of Neurology, University of California San Francisco, San Francisco, CA 94158, USA. [2]Systems Biology and Computer Science Program, Ann Romney Center for Neurological Diseases, Department of Neurology, and Division of Genetics, Department of Medicine, Brigham & Women's Hospital, Harvard Medical School, Boston, MA 02115, USA. [3]Broad Institute of Harvard University and Massachusetts Institute of Technology, Cambridge, MA 02142, USA. [4]Departments of Neurology, Yale School of Medicine, 300 George St, New Haven, CT 06511, USA. [5]Department of Genetics, Yale School of Medicine, 300 George St, New Haven, CT 06511, USA. [6]Institute of Clinical Medicine, University of Oslo, Oslo 0318, Norway. [7]Department of Neurology, Oslo University Hospital, Oslo 0424, Norway. [8]John P. Hussman Institute for Human Genomics, University of Miami, Miller School of Medicine, Miami, FL 33136, USA. [9]Dr. John T. Macdonald Foundation Department of Human Genetics, University of Miami, Miller School of Medicine, Miami, FL 33136, USA. [10]Department of Neurology, Klinikum rechts der Isar, School of Medicine, Technical University of Munich, 81675 Munich, Germany. [11]Munich Cluster for Systems Neurology (SyNergy), 81377 Munich, Germany. [12]Division of Epidemiology, School of Public HealthUniversity of California, 324 Stanley Hall, MC#3220, Berkeley, CA 94720, USA. [13]Department of Mechanical, Electronics and Chemical Engineering, Oslo Metropolitan University, Oslo 0167, Norway. [14]Section of Biostatistics, Neurophyisiology and Psychiatry, Unit of medical and genomic statistics, Universita di Pavia, Pavia 27100, Italy. [15]Department of Biomedical Sciences for Health, University of Milan, Milan 20133, Italy. [16]MS Research Unit and Department of Neurology, IRCCS Policlinico San Donato, San Donato Milanese, Milan 20097, Italy. [17]Faculty of Medicine, Westmead Clinical School, The Westmead Institute for Medical Research, Sydney, NSW 2145, Australia. [18]Department of Quantitative and Population Health Sciences, School of Medicine, Case Western Reserve University, Cleveland, OH 44106, USA. [19]Institute of Health and Society, University of Oslo, Oslo 0318, Norway. [20]Servei de Neurologia-Neuroimmunologia. Centre d'Esclerosi Múltiple de Catalunya (Cemcat), Institut de Recerca Vall d'Hebron (VHIR), Hospital Universitari Vall d'Hebron, Universitat Autònoma de Barcelona, Barcelona 08035, Spain. [21]Department of Neurology, San Raffaele Scientific Institute, Milan 20132, Italy. [22]Department of Health Sciences, UPO University, Novara 28100, Italy. [23]Department of Clinical Neurosciences. Neurology Unit, University of Cambridge, Cambridge CB2 1QW, UK. [24]Faculté de médecine, MS Clinic Centre Hospitalier de l', Université de Montréal. Université de Montréal Montreal, Montreal QC H3A 1G1, Canada. [25]Department of Neurology, Laboratory of Neurogenetics, University of Thessaly, University Hospital of Larissa, Larissa 41223, Greece. [26]Department of Neurology, University Hopital Pitié-Salpêtrière, Paris 75013, France. [27]UMR 1127, Sorbonne-Université, INSERM, University Hopital Pitié-Salpêtrière, Paris 75013, France. [28]KU Leuven Department of Neurosciences, Laboratory for Neuroimmunology, Leuven 3000, Belgium. [29]Université de Nantes, INSERM, Centre de Recherche en Transplantation et Immunologie, UMR 1064, ATIP-Avenir, Equipe 5, Nantes F-44093, France. [30]CHU de Nantes, INSERM, CIC 1413, Pôle Hospitalo-Universitaire 11: Santé Publique, Clinique des données, Nantes F-44093, France. [31]Department of Neurology, Medical School, University of Cyprus, Nicosia 587G+X2, Cyprus. [32]Institute for Science & Technology

in Medicine, Keele University, Keele ST5 5GB, UK. [33]Department of Neurology, Erasmus MC Dr Molewaterplein 40, Rotterdam 3015 GD, The Netherlands. [34]Department of Immunology, Erasmus MC, Rotterdam 3015 GD, The Netherlands. [35]First Faculty of Medicine, Department of Neurology and Center of Clinical Neuroscience, Charles University and General University Hospital, Prague 3CFG+RJ, Czech Republic. [36]Department of Neurology, Kyushu University, Kyushu 812-0053, Japan. [37]Royal Stoke MS Centre of Excellence, University Hospital North Midlands, Stoke-on-Trent, ST4 6QG, UK. [38]Department of Neurology, Medical University of Graz, Graz A-8036, Austria. [39]Department of Clinical Neuroscience and Center for Molecular Medicine, Karolinska Institutet, Stockholm 17176, Sweden. [40]Department of Neurology, University Medical Center of the Johannes Gutenberg University Mainz, Mainz 55131, Germany. [41]Genetic and Molecular Epidemiology Group, Lübeck Interdisciplinary Platform for Genome Analytics, Institutes of Neurogenetics and Cardiogenetics, University of Lübeck, Lübeck 23562, Germany. [42]Neuroimmunology and MS Research (nims), Department of Neurology, University Zurich, Zürich 8006, Switzerland. [43]Department of Neurology, section 2082, Rigshospitalet, Danish Multiple Sclerosis Center, University of Copenhagen, Copenhagen 2100, Denmark. [44]Institute for Molecular Medicine Finland (FIMM), University of Helsinki, Helsinki FIN-00014, Finland. [45]Harvard University, Center for Human Genetic Research, Boston, MA 02115, USA. [46]UK Dementia Research Institute at University College London, London WC1E6BT, UK. [47]Menzies Institute for Medical Research, University of Tasmania, Hobart, TAS 7000, Australia. [48]Department of Clinical Neurosciences, Cambridge Biomedical Campus, University of Cambridge, Cambridge CB2 0QQ, UK. [49]Department of Neurology, Center for Translational and Computational Neuroimmunology and Multiple Sclerosis Center, Columbia University Medical Center, New York, NY 10032, USA. [50]Bakar Institute for Computational Health Science. University of California San Francisco, San Francisco, CA 94158, USA

