## [peer review file · Nature Communications]

Reviewer #1 (Remarks to the Author):

The authors describe a method to better understand variants associated to Multiple Sclerosis, by ascertaining GWAS data, predict the regulatory effect of MS SNPs and using this in conjunction with cell-type specific protein-protein interaction networks.

I think I understand the approach, but unfortunately I found it a bit difficult to get an understanding how the methodology is implemented, due to the fact I found essential details lacking (see below). While I might have missed this, I believe it would be important to provide those details, in order to be ascertain the robustness of the reported associations.

Major comments:

1) The manuscript is very difficult to comprehend, and requires going back and forth to many different places in the manuscript in order to get a sense of the type of analyses conducted. This makes it hard for me to understand the strategy and to understand the novelty of the approach and the novelty of the findings with regards to multiple sclerosis. Essential details are missing, precluding me from making a fair judgment on the methodological aspects of this paper. A few of these missing details are described below:

1) The authors use epigenetic data to predict regulatory elements (PRE). However, to me it is unclear how the epigenetic annotation data from ENCODE and the Epigenome Roadmap can be immediately assigned to individual genes. How did the authors do this? E.g. let's suppose a SNP maps in a place that has a predicted repressive role, is this repressive role on the closest mapping gene? The authors also use ImmVar eQTL data, that permits empirical linking SNPs to genes, but I do not know how this could be done for the epigenetic mark data unequivocally. Can the authors elaborate on this?

2) The authors describe the 'weighted weights' (line number 406), and I do not really understand what it meant here.

3) How did the authors account for LD? E.g. how was dealt with the HLA region that has strong and extensive LD. Did the authors explicitly account their analysis for LD? What is the consequence of not having corrected for LD for loci where multiple genes map that have the same biological function? Did the authors in the network analysis ensure that genes that map in the same locus and that are PPI are not counted?

3) The authors ascertain significance by contrasting the significance score to scores obtained using 10,000 randomly generated networks. The authors however do not describe how these randomly generated networks look like: Did the authors ensure that the permuted networks have the same

degree distribution? Did the authors ensure that in for each of the 10,000 generated networks, each genes has the same number of protein-protein interactions as that particular gene has in the real network? We have observed that when these criteria are not met, many false-positive associations show an, and that the aforementioned conditions, imposed on the permuted networks, are essential in order to calibrate statistical methods that employ networks. The reason for this is that particularly with PPI networks, there are a lot of genes that do not have any established PPI, whereas a few genes have many PPIs. This number of PPIs per gene is correlated with other properties (e.g. mean gene expression levels) as well. Can the authors be explicit how these permuted networks have been generated and what kind of constraints have been used in order to make these permuted networks as similar as possible to the real networks?

4) The authors do not describe how the cell-type specific protein-protein interaction networks are actually made. The authors state they start off with a 15,783 nodes and 455,321 edges, and subsequently use gene expression values from elsewhere to create cell-type specific sub-networks. I wonder however, how the authors did this.

Did the authors consider using the Genets approach from the Lage lab? That methodology seems to resolve most of the issues described above. I would be curious to hear how the proposed method differs from the Genets method, and what the innovations of this new method are above the Genets approach.

Reviewer #2 (Remarks to the Author):

The manuscript from the IMSGC describes a novel systems biology approach to identify the cell-specific target genes from gwas summary statistics. They used the latest GWAS on multiple sclerosis (MS) to test the method and then they ran individual and cell-specific level genetic risk scores to identify individuals at high risk grouped on four main clusters: genes acting via monocyte, b cell, T cell and CNS specific effects. I find this paper extremely relevant as it could address key questions such as: target genes, target cell-types, target pathways and personalized medicine. I honestly found it hard to read, and while I deeply value the relevance of the work, I feel the paper needs to be written in a way to make it easier to follow for the Nat comm audience. Providing more detailed results for individuals that have high risk score for the different cell-types would also be appealing. My specific comments below.

Major:

The authors need to decide if they are writing a paper on MS or a paper on a new method. Regardless of that, I feel there is too much technical detail on the Results section. Nat Comm is an online journal, there is no point in repeating methods on the results like: "This resulted in nearly half a million SNPs which were then used as input in an automated query into Regulomedb

(<http://regulome.stanford.edu>) to retrieve all available regulatory information from the ENCODE and Roadmap Epigenomic Projects (Supplementary Table S1).

It is not clear from the manuscript how the authors went from regulatory features to target gene. They only say there was a “master table”. Regulomedb does not have information linking regulatory regions to genes. This is a critical step that needs to be fully described. This master table should also be posted as supplementary information if someone wants to reproduce these results.

Page 9, line 153. The authors refer to a human protein network. Checking the reference, it seems to me that this network is partially composed by human data plus data coming from yeast and other organisms. It's unclear how much of that network was constructed with data coming from CNS tissue. This could have an impact on the observations described in line 164.

Table 1. Add the pvalues of all correlation coefficients.

Page 12, line 238. Add supplementary table with pvalues for the 200 snps as tested on this 2370 patients and 412 controls. It's very likely that the study is underpowered to cluster results. What is the pvalue of predicted expression of gene vs case status?

Page 14. I feel there are too many figures and too little actual data to support the claims that cases have more interactions than controls. Instead of figures I'd like to see actual data in tables with proper statistics and distributions of number of nodes, interactions, topologies, etc.

Page 15. I really enjoyed the idea of having cell-specific genetic risk scores, but I'd like to know how much that differs from that individual total genetic risk score. A histogram showing overlapping total genetic risk score and cell-specific genetic risk score would be helpful as well as descriptive statistics.

Discussion: In their discussion, the authors speculate that these cell-specific risk scores could be useful for personalized approaches to therapy. They do not support this statement with the data shown. What is the current predictive power of these scores? Were the subjects of the validation dataset included on the main IMSCG meta-analysis?, that would overestimate the prediction ability. I would suggest to run these cell-specific scores on MS individuals who also have deep clinical/phenotypic information that could support the statement of personalized medicine.

Can the authors put their results into perspective of the “omnigenic” model proposed by Jonathan Pritchard?

Minor:

Spell out GS, SR and NR.

Supplementary tables have not clear sort. Table headers are not self-explanatory.

Authors claim this method can be implemented for other traits/diseases. Not clear how much work that would imply, it seems to be there were huge computational resources involved, ie. Can they run it for all publicly available GWAS summary stats?

Reviewer #3 (Remarks to the Author):

The authors have undertaken a number of cell-specific pathway analyses using GWAS data from studies in MS. They take both population and individual approaches - of particular interest is that it may be possible to characterise individuals according to cell-specific gene scores to allow the best choice of therapeutic. I found the paper to be fascinating. It is very well written and easy to follow even though it is presenting complex information. There are a number of minor punctuation errors that should be attended to.

Abstract

1. Nicely written. The semi-colon at line 23 would be better as a comma.

Introduction

1. Page 4, line 68 – MS is both inflammatory and neurodegenerative.
2. Very nice introduction

Results

1. Page 6. The next Supp Table mentioned after Supp Table 1 is Supp Table 5. It would make more sense for this to be Supp 2. (similarly the sequence following – next table is Supp Table 8)

2. Line 127, page 7. “in the CNS and L” – what is ‘L’? I see that this is defined in the methods, but it may need to be defined in the results, since they come before the methods (also other abbreviations used in the methods)
3. Page 10, line 193 – ‘genome-wide GW’ – are both needed?
4. Sentence starting on line 290, page 15 does not make sense.

Reviewers' comments:

Reviewer #1 (Remarks to the Author):

Major comments:

The manuscript is very difficult to comprehend, and requires going back and forth to many different places in the manuscript in order to get a sense of the type of analyses conducted. This makes it hard for me to understand the strategy and to understand the novelty of the approach and the novelty of the findings with regards to multiple sclerosis. Essential details are missing, precluding me from making a fair judgment on the methodological aspects of this paper.

We thank this reviewer for his candid opinion and advice. We have taken every possible step to clarify the message of the manuscript and have re-written large portions of it to improve its legibility.

1) The authors use epigenetic data to predict regulatory elements (PRE). However, to me it is unclear how the epigenetic annotation data from ENCODE and the Epigenome Roadmap can be immediately assigned to individual genes. How did the authors do this? E.g. let's suppose a SNP maps in a place that has a predicted repressive role, is this repressive role on the closest mapping gene? The authors also use ImmVar eQTL data, that permits empirical linking SNPs to genes, but I do not know how this could be done for the epigenetic mark data unequivocally. Can the authors elaborate on this?

We acknowledge this aspect may have not been explained in sufficient detail. The regulatory annotations mapping to a given SNP are not assigned directly to a single gene. We recognize that while a single association (top significant SNP) per locus is usually reported in a GWAS, many other variants that are in linkage disequilibrium are also likely to have an influence on the expression of nearby genes. Therefore, we integrate the potential regulatory effect of all the variants in the vicinity of the reported SNP (at different levels of LD, hence the $r^2=0.8$, $r^2=0.5$ and $r^2=0.1$ thresholds throughout the paper). Since different cell types have a different set of regulatory features, it is possible (and quite frequently observed here) that the same genomic association can have different effects on nearby genes in different cell types. With respect to the specific example this reviewer poses, the score assigned to nearby genes does not depend solely on the closest SNP, but it is a weighted sum of the effects of all SNPs in the association block under consideration. In a simple example, if only two genes (genes 1 and 2) map nearby an associated SNP (A), and there is only one additional variant (B) in close LD, then both genes would get a score composed of the weighted sum of the regulatory features annotated to each of these variants. If 10 independent experiments (reported in ENCODE or REP) describe that SNP-A maps within the promoter of gene 1, and only one experiment describes that this same SNP actually sits in a repressor for the same gene (thus contradicting the previous evidence), we consider that the likelihood that this SNP influences transcription is much higher than the chance that influences repression, and assign a positive score (+9) to this SNP-gene relationship. We then proceed to score the other SNP-gene pairs and compute the aggregate (sum) score for each gene.

Of course, in reality there are several SNPs in LD with the top association (depending on the LD threshold, up to ~100), and there are several genes that map to a given association (depending on the complexity of the genomic region, up to several dozen).

We attempted to explain this process in Figure 1b, but have now added more details to the methods section.

2) The authors describe the 'weighted weights' (line number 406), and I do not really understand what it meant here.

The weighted weights concept refers to the procedure described just above and its name derives from the fact that the sum of the effects of neighboring SNP to a given gene is weighted twice. The first time we weight the number of experiments reported in ENCODE or REP for a given SNP-gene pair (in the example above, we assign more value to a relationship that has been reported in 10 independent experiments, to another that has been reported just once). The second time, we weight the evidence stemming from all SNPs nearby a gene (depending on the LD structure there could be ~100 SNP near a given gene). If the weighted sum of the 2 SNPs near gene 1 (following the example from above) is positive (both SNPs are predicted to influence transcription) then the final score is positive. In contrast, if annotations for SNP1 suggest activation of transcription (with a weighted score of 9), but annotations for SNP2 suggest repression (with a weighted score of 20), then the net weight for that gene is -11.

3) How did the authors account for LD? E.g. how was dealt with the HLA region that has strong and extensive LD. Did the authors explicitly account their analysis for LD? What is the consequence of not having corrected for LD for loci where multiple genes map that have the same biological function? Did the authors in the network analysis ensure that genes that map in the same locus and that are PPI are not counted?

LD was definitely accounted for. Since there is no established threshold to report LD, we chose three cut-off values ($r^2=0.8$, $r^2=0.5$, and $r^2=0.1$), thus spanning a wide range of possibilities. Since the entire analysis was repeated at these three LD thresholds, the interested reader can focus on a particular cut-off or compare the results of all three.

Given its complex LD structure and its extensive characterization, we specifically excluded the HLA region from this analysis.

With respect to the last question, we searched for the possibility that genes mapping to the same locus would show more interactions than genes that mapped to different locations, and our results were reassuring. Specifically, we colored each node in the networks we report in Figure 3 by its chromosomal position and observed no clustering of genes by locus (Figure a below). We are happy to include this figure as a supplementary information if this reviewer considers it would be an important addition to the paper.

4) The authors ascertain significance by contrasting the significance score to scores obtained using 10,000 randomly generated networks. The authors however do not describe how these randomly generated networks look like: Did the authors ensure that the permuted networks have the same degree distribution? Did the authors ensure that in for each of the 10,000 generated networks, each genes has the same number of protein-protein interactions as that particular gene has in the real network? We have observed that when these criteria are not met, many false-positive associations show an, and that the aforementioned conditions, imposed on the permuted networks, are essential in order to calibrate statistical methods that employ networks. The reason for this is that particularly with PPI networks, there are a lot of genes that do not have any

established PPI, whereas a few genes have many PPIs. This number of PPIs per gene is correlated with other properties (e.g. mean gene expression levels) as well. Can the authors be explicit how these permuted networks have been generated and what kind of constraints have been used in order to make these permuted networks as similar as possible to the real networks?

We thank this reviewer for an insightful set of questions regarding the permuted networks. We wish to clarify that the “random networks” used as a background distribution were actually sampled from the exact same protein interactome as the reported networks. Thus, while nodes are selected at random, edges are preserved as they were in the original PPI therefore, if any discrepancy exists between the real and random nets is purely biological and not an artifact of the selection process. We agree that in the case of generating truly random networks (nodes and edges), care should be taken to ensure the properties of the random networks are not significantly different than that of the real nets. Our algorithm for the background distribution involves selecting a random set of nodes of equal size to that of the real network, and computing how many edges (real protein interactions) exist among them. Thus, while the nodes are selected at random, the edges are not.

To illustrate this point further, we computed additional metrics for both random networks and real networks (e.g. degree, transitivity, closeness and betweenness) and tested whether they were significantly different. As shown in Table a below (please zoom in to see details), our results demonstrate that there is no significant difference in the degree of the real and random networks (column Y). However, a significant difference can be observed in the total number of edges, the size of the largest connected component (LCC), and the number of edges in the LCC, particularly for the networks generated from genome-wide (GW) significant associations. The difference in size of the LCC is revealing as it indicates that while nodes from random nets are dispersed throughout the network, those from the real net are connected to other genes in the neighborhood. This is more formally addressed by the observed significant closeness centrality (column AA). Other metrics such as transitivity and betweenness are not significant between random and real networks, highlighting that the comparisons are made across sets of similar networks. We conclude by stating that the resulting background distribution is a valid ensemble against which to compute enrichment of edges in a biologically plausible network. We have now clarified our approach in the Methods section.

5) The authors do not describe how the cell-type specific protein-protein interaction networks are actually made. The authors state they start off with a 15,783 nodes and 455,321 edges, and subsequently use gene expression values from elsewhere to create cell-type specific sub-networks. I wonder however, how the authors did this.

For this study, we used the interactome described in *Genome research* 21, 1109-21 (2011) which comprises 15,783 nodes and 455,321 edges. Naturally, these interactions were not described in a cell specific manner, but rather are the result of multiple experiments with different cell types, thus providing a general set of interactions. In order to create cell-specific networks, we restricted interactions among gene products known to be expressed in a specific cell type (as determined by RNAseq performed in multiple tissues/cell types and described in *Sci Rep* 6, 35241 (2016)) This pruning resulted in somewhat overlapping, but distinct set of interactions for each of the cell types analyzed in this paper.

6) Did the authors consider using the Genets approach from the Lage lab? That methodology seems to resolve most of the issues described above. I would be curious to hear how the proposed method differs from the Genets method, and what the innovations of this new method are above the Genets approach.

Genets is a valuable tool that performs some of the analyses reported in the second phase of our approach, featuring an attractive user interface. However, a fundamental difference between Genets and our approach is how putative genes are prioritized (aka, the SNP-to-gene problem), a strategy that is critical for our analysis. While Genets uses associations as defined by the GWAS authors (usually based on proximity to the lead SNP), we employ an elaborated pipeline that takes into account regulatory elements (from ENCODE and REP), ultimately resulting in data-driven cell-specific pathways. In addition, our approach allows for prioritization of genes within an association block, thus affording comparisons of risk profiles in different cell types.

Reviewer #2 (Remarks to the Author):

Major:

The authors need to decide if they are writing a paper on MS or a paper on a new method. Regardless of that, I feel there is too much technical detail on the Results section. Nat Comm is an online journal, there is no point in repeating methods on the results like: “This resulted in nearly half a million SNPs which were then used as input in an automated query into Regulomedb (<http://regulome.stanford.edu>) to retrieve all available regulatory information from the ENCODE and Roadmap Epigenomic Projects (Supplementary Table S1).

We thank this reviewer for this comment and suggestion. We have now re-written large parts of the results to avoid repetition of what is included in Methods and to ensure a clear message is delivered. These changes resulted in an expanded online methods section, so the interested reader could find additional relevant information that may help interpret all our findings.

It is not clear from the manuscript how the authors went from regulatory features to target gene. They only say there was a “master table”. Regulomedb does not have information linking regulatory regions to genes. This is a critical step that needs to be fully described. This master table should also be posted as supplementary information if someone wants to reproduce these results.

We appreciate this comment, and acknowledge that our description of this part of the method may have been sub-optimal. To address this reviewer’s concern, we provide a detailed explanation below.

The regulatory annotations mapping to a given SNP are not assigned directly to a single gene. We assume that, while a single associated (top significant) SNP is usually reported in a GWAS, many other variants that are in linkage disequilibrium will also have an influence on the expression of nearby genes. Therefore, we integrate the potential regulatory effect of all the variants in the vicinity of the reported SNP (at different levels of LD, hence the $r^2=0.8$, $r^2=0.5$ and $r^2=0.1$ thresholds throughout the paper). Since different cell types have a different set of regulatory features, it is possible (and quite frequently observed here) that the same genomic association have different effects on nearby genes in different cell types. With respect to the

specific example this reviewer poses, the score assigned to nearby genes does not depend solely on the closest SNP, but it is a weighted sum of the effects of all SNPs in the LD block under consideration. In a simple example, if only two genes (genes 1 and 2) map nearby to an associated SNP (A), and there is only additional variant (B) in close LD, then both genes would get a score composed of the weighted sum of the regulatory features annotated to each variant. If 10 independent experiments (reported in ENCODE or REP) describe that SNP-A maps within the promoter of gene 1, and only one experiment describes that this same SNP actually sits in a repressor for the same gene (thus contradicting the previous evidence), we consider that the likelihood that this SNP influences transcription is much higher than the chance that influences repression, and assign a positive score (+9) to this SNP-gene relationship. We then proceed to score the other SNP-gene pairs and compute the aggregate (sum) score for each gene. Of course, in reality there are several SNPs in LD with the top association (depending on the LD threshold, up to ~100), and there are several genes that map to a given association (depending on the complexity of the genomic region, up to several dozen).

We attempted to explain this process in Figure 1b, but have now added more details to the methods section.

The “master table” aggregates all regulatory features for all SNPs of all genes. This is a very large table containing thousands of columns and half a million rows, and we deemed would not be practical for most reader to download (a table this big cannot be visualized in a spreadsheet, but only via scripts). In any case, we now make this table available as a supplementary dataset for the interested reader.

Page 9, line 153. The authors refer to a human protein network. Checking the reference, it seems to me that this network is partially composed by human data plus data coming from yeast and other organisms. It’s unclear how much of that network was constructed with data coming from CNS tissue. This could have an impact on the observations described in line 164.

The study where HumanNet is derived from is based on 18,714 human Entrez genes with validated coding proteins (downloaded from NCBI). Authors used only annotations supported by experimental evidence: IDA (inferred from direct assay); IMP (inferred from mutant phenotype); IPI (inferred from protein interaction); IGI (inferred from genetic interaction); and TAS (traceable author statement). In the construction of HumanNet, authors incorporated diverse expression, protein interaction, genetic interaction, sequence, and literature. Comparative genomics data, including both data collected directly from human genes, as well as that from orthologous genes of yeast, worm, and fly was also used to increase confidence of interactions. This network includes interactions determined by using many different cell types, but the network itself is an aggregate of all results, not cell specific. In order to create cell-specific nets, we filtered interactions realized only by gene products expressed in a given cell type, by using RNAseq expression profiles from Kitsak et al. *Sci Rep* 6, 35241 (2016). Thus, for the T cell interactome, we only retrieved interactions between proteins known to be expressed by any T cell subset present in Kitsak et al. In the case of CNS, while gene expression data is sufficiently granular (profiles for different brain cell types and regions exist), epigenomic data for CNS cells/tissues in ENCODE or REP is very sparse, thus we decided to merge all data into a single CNS category. We recognize this is a shortcoming of this study and acknowledge this as a caveat in the discussion.

Table 1. Add the p-values of all correlation coefficients.

Done

Page 12, line 238. Add supplementary table with pvalues for the 200 snps as tested on this 2370 patients and 412 controls. It's very likely that the study is underpowered to cluster results. What is the pvalue of predicted expression of gene vs case status?

As requested by this reviewer, we have conducted a GWAS of the UCSF dataset (2370 patients and 412 controls) (See Manhattan plot in Figure b below). As expected for the relatively small size of this cohort, not many significant associations were detected. However, the presence of a peak in Chromosome 6 is indicative of the known HLA association, and thus serves as validation of our analysis. Given the different platforms used for genotyping, not every SNP reported in the IMSGC GWAS was identified in this sub-analysis (specifically 148 of 200). We added this information in a new Supplementary Table (S30). Also, despite the significant correlation between PRE and gene expression, no significant clustering of patients and controls was produced when using the PREs over all genes. This is expected due to the modest (despite highly significant) correlations.

Page 14. I feel there are too many figures and too little actual data to support the claims that cases have more interactions than controls. Instead of figures I'd like to see actual data in tables with proper statistics and distributions of number of nodes, interactions, topologies, etc.

We have now added network properties corresponding to the high-level summary data presented in Supplementary Figures 1 and 2. This information is in Supplementary Table 29 and 31

Page 15. I really enjoyed the idea of having cell-specific genetic risk scores, but I'd like to know how much that differs from that individual total genetic risk score. A histogram showing overlapping total genetic risk score and cell-specific genetic risk score would be helpful as well as descriptive statistics.

We computed the polygenic risk score as previously reported (Gourraud, P.A. et al. *Annals of neurology* 69, 65-74 (2011)) and computed correlations with cell-specific PRE for all 2370 UCSF patients. The total polygenic risk score (MSGB) is significantly correlated with cell-specific genetic score (PRE) for all cells analyzed. This suggests that the PRE captures some of the global genetic risk but can differ across cell types. We have now added this analysis to Supplementary Figure S3.

Discussion: In their discussion, the authors speculate that these cell-specific risk scores could be useful for personalized approaches to therapy. They do not support this statement with the data shown. What is the current predictive power of these scores? Were the subjects of the validation dataset included on the main IMSGC meta-analysis?, that would overestimate the prediction ability. I would suggest to run these cell-specific scores on MS individuals who also have deep clinical/phenotypic information that could support the statement of personalized medicine.

Our statement was indeed speculative, but arguably posing a realistic scenario. One of the challenges of GWAS is that genotypes are easier to obtain than phenotypes, thus preventing the detailed analysis of endophenotypes. While we do have detailed clinical observations in hundreds of our patients, we are still severely underpowered to conduct any meaningful

association study. We have now clarified our statement and reads: “We speculate that in the near future this information could also be used as the basis to develop individualized risk scores, or to derive personalized approaches to therapy.”

Can the authors put their results into perspective of the “omnigenic” model proposed by Jonathan Pritchard?

Indeed, our data is in agreement with this model, which proposes that gene regulatory networks are sufficiently interconnected such that all genes expressed in disease-relevant cells are liable to affect the functions of core disease-related genes. In fact, authors of that paper mention that “Autoimmune GWAS hits affect shared and tissue-specific regulation of immune cells”, which we demonstrate in Figure 3. We now discuss this relevant study in the main text and provide the corresponding reference.

Minor:

Spell out GS, SR and NR.

These are spelled out at the beginning of the Online Methods section. The paragraph reads: “Specifically, single nucleotide polymorphisms (SNP) corresponding to all non-MHC genome-wide (GW) (n=200; Supplementary Table S2), statistically replicated effects (SR) (n=416; Supplementary Table S3) and non-replicated effects (NR) (n=3695; Supplementary Table S4) were extracted for analysis GW, SR and NR as defined previously.”

Supplementary tables have not clear sort. Table headers are not self-explanatory.

A key has been added below each Supplementary Table describing the meaning of each header.

Authors claim this method can be implemented for other traits/diseases. Not clear how much work that would imply, it seems to be there were huge computational resources involved, ie. Can they run it for all publicly available GWAS summary stats?

This reviewer is correct in that significant computational resources are required to conduct this analysis. We are currently working on a web-based solution to enable any interested investigator to paste summary-level GWAS data in a query box and submit the job, which would be computed in our cluster. The researcher can then download a set of files with the network results, statistics, and visualizations.

Reviewer #3 (Remarks to the Author):

Abstract

Nicely written. The semi-colon at line 23 would be better as a comma.

We have replaced the semi-colon by a comma.

Introduction

Page 4, line 68 – MS is both inflammatory and neurodegenerative.

We have now added the suggested language

Very nice introduction

Thank you!

Results

1. Page 6. The next Supp Table mentioned after Supp Table 1 is Supp Table 5. It would make more sense for this to be Supp 2. (similarly the sequence following – next table is Supp Table 8)

We have corrected these discrepancies and all the supplementary tables are in logical order now.

2. Line 127, page 7. “in the CNS and L” – what is ‘L’? I see that this is defined in the methods, but it may need to be defined in the results, since they come before the methods (also other abbreviations used in the methods)

We have defined L (lung) upon first mention.

3. Page 10, line 193 – ‘genome-wide GW’ – are both needed?

Thank you for catching this error. We have now corrected it.

4. Sentence starting on line 290, page 15 does not make sense.

We have now re-written this sentence. It now reads: “Another interesting observation emerging from this analysis is that subjects at the extremes of the distribution of intra-individual interactions (a proxy for their overall risk) can be identified for each cell type.”

Appendix:

Figure a. Genes in reported networks are not affected by linkage disequilibrium. Each gene is colored based on its chromosomal position (cytogenetic band). No clustering of genes with the same color can be observed indicating there is no enrichment of interactions due to chromosomal vicinity in these networks.

Figure b. Mahattan plot of GWAS of UCSF dataset (2370 patients and 412 controls).

Significance	CellType	Qst	Threshold	quante	Threshold	Nodes	Edges	LCC nodes	LCC edges	Degree	Transitivity	Clustering	Betweenness	indegrees rand	nodes LCC rand	edges LCC rand	randDegree	randTransitivity	randClustering	randBetweenness	randInBtw	eval_indegrees	eval_nodes LCC	eval_edges LCC	eval_Degree	eval_Transitivity	eval_Clustering	eval_Betweenness
GW	B	0.5	0.1	0.25	110	115	67	114	2.09	0.347	0.0015	62.3	65.7	37.9	37.9	9.55	0.402	9.3E-05	54.0	2255	4.3E-03	2.9E-03	5.2E-03	0.48	0.81	6.03E-02	0.41	
GW	B	0.5	0.25	0.89	106	111	65	111	2.09	0.251	0.0017	60.8	61.4	35.6	53.0	9.46	0.405	9.8E-05	51.5	2166	2.3E-03	2.3E-03	2.8E-03	0.48	0.80	3.0E-02	0.40	
GW	B	0.5	0.5	2.00	76	35	22	28	0.92	0.162	0.00020	6.7	31.3	18.2	23.8	10.35	0.420	1.6E-04	37.2	1791	0.3600	0.3150	0.377	0.70	0.91	0.190	0.81	
GW	T	0.5	0.1	0.25	99	80	48	74	1.63	0.263	0.00215	50.7	41.8	22.3	31.5	5.42	0.400	1.0E-04	27.5	2832	6.07E-03	1.74E-03	1.95E-03	0.43	0.77	3.0E-04	0.17	
GW	T	0.5	0.25	0.89	95	75	48	70	1.58	0.233	0.00216	51.2	40.2	20.3	29.2	5.41	0.403	1.1E-04	25.7	2489	5.80E-03	7.36E-04	6.12E-03	0.43	0.82	1.12E-04	0.14	
GW	T	0.5	0.5	2.00	80	55	30	47	1.38	0.212	0.00219	10.9	28.5	14.4	19.4	5.58	0.403	1.4E-04	20.7	2366	6.48E-03	1.13E-02	1.13E-02	0.44	0.84	1.48E-02	0.67	
GW	M	0.5	0.1	0.25	70	45	21	36	1.29	0.463	0.0024	8.2	28.5	10.0	13.7	4.47	0.407	1.8E-04	17.1	2050	7.23E-04	6.08E-03	1.13E-03	0.46	0.40	2.02E-02	0.68	
GW	M	0.5	0.25	0.89	68	43	21	34	1.26	0.500	0.00205	6.8	19.3	9.5	10.9	4.52	0.403	1.9E-04	16.8	2014	6.77E-04	1.00E-02	6.72E-04	0.46	0.33	2.70E-02	0.70	
SR	B	0.5	0.1	0.25	203	203	159	193	1.98	0.264	0.00203	80.1	213.7	108.1	217.0	8.71	0.371	3.8E-05	115.1	2187	4.77E-01	4.04E-01	0.761	0.76	0.8910	0.83	0.76	0.8910
SR	B	0.5	0.25	0.89	193	182	96	170	1.89	0.245	0.00204	77.4	202.7	97.0	194.4	8.77	0.376	4.0E-05	105.9	2498	0.6930	0.5350	0.715	0.70	0.80	0.5610	0.77	
SR	B	0.5	0.5	2.00	117	62	39	48	1.66	0.211	0.00209	22.3	74.7	42.8	65.5	9.12	0.400	8.4E-05	58.0	1335	0.7340	0.6380	0.748	0.70	0.80	0.4750	0.84	
SR	T	0.5	0.1	0.25	167	113	67	109	1.81	0.260	0.00205	58.7	124.9	61.3	111.4	5.14	0.399	4.4E-05	65.0	3687	0.6520	0.5920	0.691	0.66	0.76	0.4330	0.58	
SR	T	0.5	0.25	0.89	158	104	64	95	1.33	0.288	0.00205	57.2	109.2	55.7	95.5	5.17	0.397	4.9E-05	58.0	3332	0.5750	0.2650	0.567	0.65	0.76	0.2020	0.51	
SR	T	0.5	0.5	2.00	109	61	36	47	1.65	0.235	0.00203	15.2	61.0	27.0	46.7	5.40	0.405	8.2E-05	51.7	3137	0.3730	0.1930	0.232	0.65	0.60	0.1420	0.74	
SR	M	0.5	0.1	0.25	181	181	99	176	2.00	0.335	0.00205	91.3	137.8	75.9	124.5	3.05	0.381	4.0E-05	83.6	4524	0.6649	0.6531	0.663	0.65	0.62	2.56E-02	0.40	
SR	M	0.5	0.25	0.89	165	165	88	159	2.00	0.359	0.00206	75.6	134.3	65.4	99.0	3.83	0.381	4.6E-05	70.7	3943	1.92E-02	4.69E-02	2.64E-02	0.61	0.55	1.33E-02	0.43	
SR	M	0.5	0.5	2.00	109	83	49	74	1.52	0.309	0.00211	32.2	59.0	25.1	34.7	4.11	0.394	8.7E-05	31.1	2913	6.66E-03	1.68E-02	1.55E-02	0.65	0.65	6.78E-03	0.52	
NR	B	0.5	0.1	0.25	176	127	67	117	1.44	0.144	0.00205	79.8	168.4	88.3	160.2	8.95	0.381	4.6E-05	84.4	1117	0.8750	0.4010	0.871	0.71	0.93	0.4390	0.65	
NR	B	0.5	0.25	0.89	157	103	69	89	1.31	0.124	0.00205	51.2	133.7	70.3	125.5	6.00	0.384	5.4E-05	62.3	1190	0.8400	0.5400	0.827	0.71	0.94	0.5170	0.60	
NR	T	0.5	0.1	0.25	128	61	40	46	0.95	0.134	0.00207	18.7	73.4	37.6	60.0	5.19	0.400	6.7E-05	41.3	2055	0.7220	0.4180	0.710	0.66	0.94	0.3660	0.80	
NR	T	0.5	0.25	0.89	115	57	37	43	0.99	0.153	0.00209	17.6	59.4	30.3	46.8	5.13	0.405	7.9E-05	34.4	2709	0.5590	0.2700	0.565	0.66	0.92	0.2100	0.75	
NR	C	0.5	0.1	0.25	81	31	21	29	0.77	0.161	0.00212	7.2	28.3	15.9	15.0	5.89	0.381	1.5E-04	11.3	3194	0.2960	0.0970	0.158	0.68	0.81	0.3020	0.67	
NR	M	0.5	0.1	0.25	142	77	56	71	1.09	0.202	0.00206	45.9	84.8	48.4	69.0	3.85	0.381	5.8E-05	51.3	2738	0.6520	0.2490	0.469	0.69	0.80	0.2790	0.61	
NR	M	0.5	0.25	0.89	122	53	31	43	0.87	0.223	0.00207	16.4	61.4	32.1	46.4	3.88	0.389	7.8E-05	40.3	2003	0.7220	0.4060	0.399	0.69	0.71	0.4440	0.62	

Table a. Characteristics of real and random networks used as background distribution.

Reviewer #1 (Remarks to the Author):

The authors have addressed various of the methodological questions that I had raised.

Although the clarity of the manuscript has improved, it remains a bit difficult to understand and appreciate the exact analytical strategy that the authors have applied.

The only thing I can suggest to help resolve this is to include a very clear workflow figure as supplemental figure, and ideally to show some comparisons with other pathway enrichment methods (e.g. FUMA / DEPICT / PASCAL) to show at least some consistency of this approach with previously and often used pathway enrichment methods.

Reviewer #2 (Remarks to the Author):

The authors have answered all my queries to my satisfaction.

REVIEWERS' COMMENTS:

Reviewer #1 (Remarks to the Author):

The authors have addressed various of the methodological questions that I had raised.

Although the clarity of the manuscript has improved, it remains a bit difficult to understand and appreciate the exact analytical strategy that the authors have applied. The only thing I can suggest to help resolve this is to include a very clear workflow figure as supplemental figure, and ideally to show some comparisons with other pathway enrichment methods (e.g. FUMA / DEPICT / PASCAL) to show at least some consistency of this approach with previously and often used pathway enrichment methods.

We have now re-analyzed our data with each of the suggested methods. Although each method is unique and produces a different output, we were able to compare the one feature that all have in common, which is the prioritization of genes within each associated locus. This additional analysis resulted in 2 new Supplementary Tables where we compare the features of each method (Supplementary Table 1) and the genes prioritized by each (Supplementary Table 2).

In addition, we created Supplementary Figure 5, where the precise overlap in gene prioritization by each method can be easily observed.

We have also created a workflow diagram of our approach (Supplementary Figure 1)